# A Comparative Study of Genetic Responses to Short- and Long-Term Habitat Fragmentation in a Distylous Herb *Hedyotis chyrsotricha* (Rubiaceae)

**DOI:** 10.3390/plants11141800

**Published:** 2022-07-07

**Authors:** Na Yuan, Shujing Wei, Hans Peter Comes, Sisheng Luo, Ruisen Lu, Yingxiong Qiu

**Affiliations:** 1Provincial Key Laboratory of Agrobiology, Institute of Crop Germplasm and Biotechnology, Jiangsu Academy of Agricultural Sciences, Nanjing 210014, China; thefuries@163.com; 2Guangdong Academy of Forestry, Guangzhou 510520, China; weishujing2003@163.com (S.W.); sisheng8009@126.com (S.L.); 3Department of Environment & Biodiversity, Salzburg University, A-5020 Salzburg, Austria; hans-peter.comes@plus.ac.at; 4Institute of Botany, Jiangsu Province and Chinese Academy of Sciences, Nanjing 210014, China; 5Wuhan Botanical Garden, Chinese Academy of Sciences, Wuhan 430074, China

**Keywords:** habitat fragmentation, RAD-Seq, *Hedyotis chrysotricha*, genetic diversity, population structure, gene flow

## Abstract

The genetic effects of habitat fragmentation are complex and are influenced by both species traits and landscape features. For plants with strong seed or pollen dispersal capabilities, the question of whether the genetic erosion of an isolated population becomes stronger or is counterbalanced by sufficient gene flow across landscapes as the timescales of fragmentation increase has been less studied. In this study, we compared the population structure and genetic diversity of a distylous herb, *Hedyotis chyrsotricha* (Rubiaceae), in two contrasting island systems of southeast China. Based on RAD-Seq data, our results showed that populations from the artificially created Thousand-Island Lake (TIL) harbored significantly higher levels of genetic diversity than those from the Holocene-dated Zhoushan Archipelago (ZA) (*π* = 0.247 vs. 0.208, *H*_O_ = 0.307 vs. 0.256, *H*_E_ = 0.228 vs. 0.190), while genetic differences between island and mainland populations were significant in neither the TIL region nor the ZA region. A certain level of population substructure was found in TIL populations, and the level of gene flow among TIL populations was also lower than in ZA populations (m = 0.019 vs. 0.027). Overall, our comparative study revealed that genetic erosion has not become much stronger for the island populations of either the TIL or ZA regions. Our results emphasized that the matrix of water in the island system may facilitate the seed (fruit) dispersal of *H. chrysotricha*, thus maintaining population connectivity and providing ongoing resilience to the effects of habitat fragmentation over thousands of years.

## 1. Introduction

Over the past 200 years, human activities have altered more than 80% of the terrestrial world, leading to increasing habitat reduction and fragmentation of almost all ecosystems [1,2,3]. Theoretically, habitat fragmentation has severe negative impacts on community structures, population distributions and abundances, patterns of genetic variation, gene flow, etc. [4,5]. These changes, especially genetic alterations, can reduce the individual fitness and evolutionary potential of populations, further impeding the long-term persistence of species and increasing the risk of local extinction [6,7].

Detecting the genetic effects of habitat fragmentation on natural populations is challenging in practice, as there are many confounding factors involved [2]. For instance, life-history traits such as life span, pollination, and seed dispersal, are essential in determining the magnitude of the plant’s response to habitat fragmentation. Short-lived herbs experience more generations and may show a faster decrease in genetic diversity than longer-lived trees or shrubs over the same timescale of fragmentation [8]. In addition, plants with abiotic-mediated pollination and seed dispersal are expected to be less susceptible to habitat fragmentation than those with animal-mediated pollination and seed dispersal [9,10]. For example, Wang et al. (2011) [11] observed no significant difference in genetic diversity between pre- and post-fragmentation cohorts of a wind-pollinated tree *Castanopsis sclerophylla* (Lindl. & Paxton) Schottky in a recently fragmented island system. Due to the disappearance of a genetic barrier, the genetic structure of post-fragmentation cohorts was significantly weakened by increased wind speeds and easier pollen movement over water. In contrast, our previous study on the distylous herb *Hedyotis chrysotricha* (Palib.) Merr in the Thousand-Island Lake (TIL) region revealed that 53 years of fragmentation has led to the loss of c. 17.4% of the initial genetic diversity in the island populations [12]. *Hedyotis chrysotricha* is insect-pollinated, with seeds that are dispersed by wind or (and) water [13,14]. Although patterns of gene flow have not been greatly modified among the *H. chrysotricha* populations, it is uncertain whether extensive seed flow can counteract the negative effects of genetic drift to reach a new dynamic equilibrium and assist the long-term persistence of remnant populations.

The various island systems in southeastern China provide natural laboratories for testing the genetic responses of plants to habitat fragmentation. For example, a long-established hotspot for testing the predictions of population genetics theory on recent anthropogenic habitat fragmentation is the TIL region in southeast China (Chun’an County, Hangzhou City, Zhejiang Province, China). This artificial lake (c. 573 km^2^), with about 1078 islands of different sizes (0.25–1320 ha) and shapes (c. 83 km^2^ in total), was formed in 1959 after the construction of the Xinanjiang River hydroelectric power station [15]. In contrast, the Zhoushan Archipelago (ZA), located in the East China Sea off the northern coast of Zhejiang Province, is an ideal candidate for examining the genetic responses to long-term habitat fragmentation. It was originally an extended part of the nearby continent and was formed by rising sea levels during the early Holocene, c. 7000–9000 years ago. It consists of 1339 islands and 3306 reefs, covering about 1440 km^2^ in total [16,17]. The two island systems differ greatly in spatial scale and temporal origin, making them ideal systems for comparative studies detecting both short-term and long-term responses of species to habitat fragmentation. For example, Yuan et al. (2015) [18] compared the patterns of nuclear microsatellite (SSR) variation and the population structure of an evergreen shrub, *Loropetalum chinense* (R. Br.) Oliver in these two island systems. The results showed that this species can mitigate the negative effects of millennia-old natural habit fragmentation via water-facilitated seed dispersal. Similar results were also observed in a comparative genetic study of the perennial vine *Actinidia chinensis* Planch. [19]. For long-lived species, the occurrence of gene flow among populations has great potential to mitigate the genetic erosion caused by both recent and historical habitat fragmentation [20]. However, similar comparative genetic studies to determine whether the same is true for short-lived, heterostylous plants, e.g., *H. chrysotricha*, are still lacking.

*Hedyotis chrysotricha* is a short-lived, subtropical forest understory herb of southeast China, with a wide distribution in both the TIL and ZA regions [21,22]. This insect-pollinated species has two floral morphs, i.e., long-styled vs. short-styled, which differ reciprocally in the placement of stigmas and anthers to enhance outcrossing. In an ideal population, an equal morph ratio is expected for the availability of compatible pollen and reproductive success [23]. However, skewed morph ratios have been observed in small and fragmented populations of many heterostylous species as a result of demographic stochasticity [24,25]. Morph types differ not only in herkogamy, but also in self- and intramorph compatibility levels, etc. This may further affect levels of inbreeding and population genetic diversity, and the ability to cope with the effects of habitat fragmentation [26,27,28]. For instance, van Rossum and Triest (2008) [29] found that pin and thrum individuals of distylous *Primula veris* differed in fine-scale spatial genetic structure patterns at a small scale, which may be explained by partial self-compatibility of the pin morph combined with restricted seed dispersal and pollinator behavior. In our previous genetic study [13], we surveyed populations of *H. chrysotricha* from the recently fragmented TIL region only, using nuclear SSR markers. Although SSRs are highly polymorphic markers, a limited number of markers might result in low power for addressing landscape genetic questions at large spatial and temporal scales [30,31,32]. With the decreasing cost of next-generation sequencing, restriction site-associated DNA sequencing (RAD-Seq), which enables the fast identification of thousands of single nucleotide polymorphisms (SNPs) in non-model organisms without any prior information, has become a cost-effective approach in phylogeographic and population genetic studies [33]. In this study, we employed genome-wide SNPs derived from RAD-Seq data to examine and compare short-term vs. long-term effects of habitat fragmentation on *H. chrysotricha* in the TIL vs. ZA regions. Specifically, we aimed to: (1) investigate the effects of short-term vs. long-term habitat fragmentation on the species’ population genetic variation (diversity, inbreeding) patterns; (2) examine the genetic differentiation and genetic structure of *H. chrysotricha* populations at small and large spatial scales; and (3) evaluate patterns of gene flow and demographic changes of *H. chrysotricha* populations in the two island systems. This comparative study will increase our understanding of the genetic responses of short-lived plants with strong seed dispersal capabilities to fragmentation and provide an insight into the management and conservation of subtropical forest dwellers in fragmented habitats at both small and large spatial–temporal scales.

## 2. Results

### 2.1. RAD-Seq Data and SNP Filtering

Approximately 389.4 billion (389,411,194,200) raw reads were produced from 256 individuals of *H. chrysotricha* sampled across the TIL and ZA regions. After quality control, filtering, and trimming, an average of 1421.5 million and 1638.9 million high-quality reads were retained for the TIL and ZA samples, respectively (Appendix A). A catalog containing 9,131,292 loci was constructed, and 9,067,359 loci were genotyped by GSTACKS using the data sets of all *H. chrysotricha* samples. The mean, minimum, and maximum values for effective per-sample coverage were 26.9×, 16.8×, and 53.0×, respectively. After filtering loci of low quality (minor allele frequency < 0.01; missing rate > 0.5), 185 and 188 unlinked SNPs were eventually identified in TIL and ZA populations, respectively, and used for all subsequent analyses (Appendix A).

### 2.2. Population Variation

Based on genome-wide SNP markers, we detected significantly higher levels of genetic diversity in *H. chrysotricha* from the TIL region than from the ZA region (*π* = 0.247 vs. 0.208, *p* < 0.01; *H*_O_ = 0.307 vs. 0.256, *p* < 0.01; *H*_E_ = 0.228 vs. 0.190, *p* < 0.01; Table 1 and Table 2). Within each region, there were no major differences in genetic diversity between island and mainland populations (TIL island vs. mainland populations: *π* = 0.257 vs. 0.237, *H*_O_ = 0.313 vs. 0.302, *H*_E_ = 0.237 vs. 0.219; ZA island vs. mainland populations: *π* = 0.208 vs. 0.207, *H*_O_ = 0.256 vs. 0.255, *H*_E_ = 0.189 vs. 0.192). *F*_IS_ values were all negative in all *H. chrysotricha* populations, with a mean value of −0.078 and −0.066 in the TIL and ZA regions, respectively (Table 1 and Table 2).

Pairwise *F*_ST_ values between the TIL populations ranged from 0.001 to 0.393, with about 13% being significant (*p* < 0.05) after sequential Bonferroni correction (Table 3). Among the significant pairwise comparisons, the great majority (11 out of 16) involved population EM04. Pairwise *F*_ST_ values between the ZA populations ranged from 0.001 to 0.436, with about 24% being significant (*p* < 0.05) after sequential Bonferroni correction (Table 4). Here, a large proportion of significantly high *F*_ST_ values (6 out of 13) involved population ZI05. The average *F*_ST_ value across populations was slightly higher in the TIL (*F*_ST_ = 0.103) region than in the ZA (*F*_ST_ = 0.092) region, yet this difference was only marginally significant (*p* = 0.046). No IBD pattern was found in either region (TIL: *r* = 0.181, *p* = 0.174; ZA: *r* = 0.072, *p* = 0.242) (Appendix A).

### 2.3. Population Genetic Structure

Based on a STRUCTURE analysis, the most likely number of genetic groups was *K* = 3 for the TIL populations and *K* = 2 for the ZA populations. In the TIL region, Cluster I (‘dark blue’) was present at high frequency (80% of all local samples) in seven of eight island populations, whereas the great majority of individuals (89%) from the western mainland populations and the remaining island population (IP08) were assigned to Cluster II (‘red’) (Figure 1 and Appendix A). In the eastern mainland populations, half of the individuals from EM04 were assigned to Cluster III (‘green’); the remaining populations contained variable numbers of individuals belonging to both Cluster I and Cluster II (Figure 1). In the ZA region, nearly all populations (excepting ZI04) consisted of variable numbers of individuals with a probability of ≥ 0.5 of belonging to both Cluster I (‘blue’) and Cluster II (‘orange’), when *K* = 2 (Figure 2). At *K* = 3–5, we also observed that the population ZM02 was assigned to a separate cluster (Appendix A), suggesting potential genetic differentiation between this mainland population and the others.

### 2.4. Contemporary Gene Flow

In *H. chrysotricha* populations from the TIL region, the proportion of individuals that originated from within the same site ranged from 68% (IP04) to 81% (WM02), with an average value of 72% (see BAYESASS, Appendix A). In contrast, rates of gene flow (or ‘migration’, *m*) between populations were low to moderate, and in the majority of populations no more than 5% of the individuals were exchanged with other populations (Appendix A). Similarly, in the ZA region, populations largely consisted of individuals originating from the same site (73% on average) (see BAYESASS, Appendix A). Nevertheless, the average rate of interpopulation gene flow in the ZA region (*m*_ZA_ = 0.027) was significantly higher than in the TIL region (*m*_TIL_ = 0.019; *p* = 0.003). The DIVMIGRATE method also suggested that the relative migration rate was significantly higher in the ZS region than in the TIL region (*rm*_ZA_ = 0.349, *rm*_TIL_ = 0.252, *p* < 0.01) (Appendix A). In addition, we observed significant asymmetric gene flow patterns in 10 out of the 16 populations in the TIL region, and 4 out of the 11 populations in the ZA region (Figure 3 and Figure 4).

### 2.5. Demographic (Bottleneck) Analyses

Irrespective of the mutation model assumed (IAM, infinite allele model; SMM, stepwise mutation model; or TPM, two-phase mutation model), we found a significant excess of heterozygosity (*p* < 0.05) for one island (IP05) and one mainland population (EM04) in the TIL region, and one island population (ZI01) in the ZA region (Table 5), indicating that these populations may have experienced a recent bottleneck. However, a significant excess of heterozygosity was only found under the IAM model for one TIL population (IP08) and five ZA populations (ZI02, ZI07, ZM02, ZM03, ZM04; Table 5).

## 3. Discussion

Habitat fragmentation changes continuous habitats into small and isolated patches, and populations inhabiting these habitats are often considered to have low genetic diversity [8,34]. In our previous study [12], we found that island populations of *H. chrysotricha* in the TIL region had significantly lower mean genetic diversity than those from the mainland, based on nuclear SSR markers, suggesting that anthropogenic habitat fragmentation can lead to significant loss of genetic diversity over a few decades. In the present study, we further tested whether genetic erosion increases with an increase in spatio-temporal scales of fragmentation. Based on RAD-Seq-derived SNP markers, we observed that the average estimates of genetic diversity for ZA populations were significantly lower than for TIL populations (*π* = 0.247 vs. 0.208, *p* < 0.01; *H*_O_ = 0.307 vs. 0.256, *p* < 0.01; *H*_E_ = 0.228 vs. 0.190, *p* < 0.01; Table 1 and Table 2), and we found no major differences in genetic diversity between island and mainland populations in either the TIL or the ZA region. Although it remains unknown how much of the initial genetic variation has been lost in each region since the formation of islands, these results may indicate that *H. chrysotricha* populations have retained a considerable proportion of their initial genetic diversity over either ~60 years or thousands of years of habitat fragmentation.

Generally, the genetic effects of habitat fragmentation on plants vary, depending on their life-history traits [2]. In this study, the lack of a significant effect of habitat fragmentation on the genetic diversity of *H. chrysotricha* may be explained by demographic factors and/or strong gene dispersal capacity. As a distylous plant species, *H. chrysotricha* has two contrasting flower morphs that reciprocally differ in the spatial separation of stigmas and anthers, which in turn prevents selfing and intramorph mating. Morph bias is expected to reinforce genetic drift effects and inbreeding by reducing mating opportunities [35,36]. We also found that no biased morph ratio existed in either island or mainland populations, and overall levels of inbreeding in both island regions were negative and close to zero, suggesting a sufficient within-population heterozygosity (Table 1 and Table 2). In addition, most of the populations showed no genetic signatures of recent bottlenecks (Table 5). Therefore, it is likely that population sizes of *H. chrysotricha* have remained sufficiently large to prevent loss of genetic diversity via inbreeding and genetic drift after fragmentation [37,38].

The gene dispersal ability via pollen or seed is another key characteristic that determines the potential of a plant species to counteract the negative effects of habitat fragmentation [8]. *Hedyotis chrysotricha* is an insect-pollinated plant; its flower visitors include solitary bees (Halictidae and Andrenidae), Diptera (Syrphidae and Bombyliidae), honey bees, small lepidopterans, and/or thrips [13,39]. Due to the limited foraging distances of these pollinators, they seem unlikely to be able to maintain significant levels of gene flow and population connectivity across fragmented habitats. In contrast, the indehiscent fruit of *H. chrysotricha* contains several very small seeds (1.5–2 × 2–2.5 mm), whose dispersal should be easily facilitated by wind [21]. In our previous study, we detected a moderate level of gene flow among 18 *H. chrysotricha* populations in the TIL region [12]. Based on high-quality SNP markers, our genetic analyses here have further verified the undisturbed population connectivity of *H. chrysotricha* populations in the TIL region, with an average of c. 28% individuals in each population being exchanged with other sites (Appendix A). Given the relatively large spatial and temporal scales of habitat fragmentation in the ZA region (Figure 5), one might expect high population differentiation and low levels of gene flow between island populations. However, both BAYESASS and DIVMIGRATE analyses suggested that the average level of gene flow among populations in the ZA region was higher than in the TIL region, although the results from these two methods may not be compared directly. Similarly, populations in the ZA region exhibited marginally lower levels of genetic differentiation than those in TIL region (*F*_ST_ = 0.092 vs. 0.103). A previous study showed that water greatly facilitated the seed dispersal of a typical wind disperser across a fragmented landscape [40]. By referencing the population genetic study of the maritime *Hedyotis strigulosa* var. *parviflora* (Hook. et Arn.) Yamazaki [14], we speculated that water flow may also facilitate the fruit dispersal of *H. chrysotricha*. Although the hydration of *H. chrysotricha* could remain speculative due to the lack of detailed morphological and experimental evidence, the genetic evidence presented here further verified that the species has sufficient fruit (seed) dispersal capabilities to maintain moderate-to-high levels of ongoing gene flow and population connectivity across fragmented landscapes at large spatial and temporal scales.

Corresponding to the above results, little evidence of strong genetic structuring was found in *H. chrysotricha* within the two island systems. Based on SNP markers, the STRUCTURE analysis grouped the TIL populations basically into three clusters (Figure 1) and the ZA populations into two clusters (Figure 2). In the TIL region, most of the island populations were genetically separated from western mainland populations, which is consistent with the results of our previous nSSR analysis, whereas the genetic composition of eastern mainland TIL populations was slightly different from the former study, consisting of a mixture of all three clusters [12]. In the ZA region, the genetic divergence between the mainland and island populations was not as high as expected, and most of the island populations still feature a high percentage of genetic admixtures, even after thousands of years of isolation (Figure 2). The higher levels of population substructure in the TIL region (three clusters) than in the ZA region (two clusters) may be attributed to their contrasting landscape histories. For example, before water flooded the TIL region, mountains (e.g., the Baiji Mountains in the western part of the TIL region) may have acted as a stronger dispersal barrier than water for gene exchange between populations inhabiting the middle and western former hilltops. In the case of the ZA region, which separated from the nearby mainland 7000–9000 years ago due to a rise in sea level, water has been instrumental in facilitating gene exchange between populations for a much longer time than in the TIL region. When combined with the lack of isolation by distance in each island system, these results further suggested that population connectivity in *H. chrysotricha* has not been greatly modified in either island system, regardless of their significant differences in spatial–temporal characteristics.

## 4. Materials and Methods

### 4.1. Sample Collection and DNA Extraction

We collected a total of 160 *H. chrysotricha* individuals from eight islands and eight adjacent mainland populations in the TIL region, and 110 individuals from seven islands and four adjacent mainland populations in the ZA region (Table 1 and Table 2; Figure 5). In each population, almost equal numbers of individuals with long-styled (L-morph) or short-styled (S-morph) flowers were collected separately. We extracted the total genomic DNA from the dry leaf material of each sample using DNA Plantzol Reagent (Invitrogen, Carlsbad, CA, USA), following the manufacturer’s protocol. We then checked the quality of the DNA using 1% agarose gel electrophoresis and measured the DNA concentration using a NanoDrop 2000 spectrophotometer (Thermo Scientific, Wilmington, DE, USA). Finally, 256 of the 270 DNA samples met the minimum quality requirements and were retained for subsequent sequencing and genotyping.

### 4.2. RAD Sequencing and Data Processing

RAD libraries were prepared using the restriction enzyme *Eco*RI and sample-specific barcodes. Samples in the libraries were pooled and sequenced on an Illumina HiSeq 2500 to generate 150 bp pair-end reads, at the Novogene Bioinformatics Institute (Beijing, China). Standard quality control (QC) pipelines were used to process the raw sequencing data. Reads with > 5% of unknown nucleotides, low-quality reads (quality value ≤ 5), and reads that did not contain sample-specific barcodes were discarded. After filtering procedures, the quality-filtered reads data for each individual were then assembled into de novo loci, and SNP calling was performed using the following core programs: USTACKS, CSTACKS, SSTACKS, GSTACKS, and POPULATIONS, implemented in the STACKS pipeline v2.2 [41]. Briefly, all sequences were firstly processed in USTACKS to produce consensus sequences of RAD tags. The program USTACKS takes a set of short-read sequences as input and aligns them into exactly matching stacks. The minimum depth of coverage required for creating a stack was set at three (–m: 3) sequences, and the maximum distance allowed between stacks was set at four (–M: 4) nucleotides. The program CSTACKS was used to build a catalog of consensus loci containing all the loci from all the *H. chrysotricha* samples and merging all alleles together, with the number of mismatches allowed between sample loci set to four (–n: 4). Next, sets of stacks (i.e., putative loci) created in USTACKS were compared against the catalog using SSTACKS. Then, GSTACKS was used to align the reads to the locus and to call SNPs. Finally, the catalog of reads was filtered using the POPULATIONS program to produce a data set for downstream analyses. To identify high-credibility SNPs, polymorphic RAD loci that were present in at least 75% of the individuals in each population were retained (–r: 0.75). Potential homologs were excluded by removing loci showing heterozygosity of >0.5 within samples [42], and one SNP per RAD locus was kept, to reduce the impact of linkage disequilibrium.

### 4.3. Genetic Diversity and Population Structure

We calculated the observed heterozygosity (*H*_O_), expected heterozygosity (*H*_E_), nucleotide diversity (π), and inbreeding coefficient (*F*_IS_) for each population using the POPULATIONS program in STACKS. Pairwise *F*-statistics (*F*_ST_) (1000 permutations) among populations were calculated using ARLEQUIN v3.5.2 [43]. Population structure was analyzed using the program STRUCTURE v2.3.5 [44]. The number of genetic clusters (*K*) was set from 1 to 16 for TIL populations and from 1 to 11 for ZA populations, corresponding to the number of sampled populations in each region. Ten independent iterations were conducted for each *K*, with a burn-in of 10,000 and 100,000 Markov chain Monte Carlo replicates, by assuming the admixture model and independent allele frequencies. The optimal *K* value was determined from the ΔK values calculated by STRUCTURE HARVESTER [45]. A Mantel test for genetic differentiation [*F*_ST_/(1 − *F*_ST_)] against geographic distance (log10 transformed) was performed in GENALEX v6.1 [46] for each region, to test the pattern of isolation by distance (IBD). Statistical significance was determined with 1000 permutations.

### 4.4. Gene Flow and Demographic History Analyses

For each region, we estimated the levels of interpopulation gene flow using the program BAYESASS v3.03 [47]. First, we ran BAYESASS with the default delta values for allelic frequency, migration rates, and inbreeding coefficients. Then, subsequent runs were adjusted with different delta values to ensure that the acceptance rate ranged between 40 and 60% for each parameter [47]. We performed 10 independent runs (1 × 10^7^ iterations with a burn-in of 10^6^ generations), each with a different initial seed. Model convergence was assessed by the program TRACER v1.5 [48]. In addition, we also used the DIVMIGRATE function from the DIVERSITY package v1.9.90 in R [49] to calculate the relative direction of the gene flow between island and mainland populations, using Nei’s *G*_ST_ method. To test for asymmetric flow (significantly higher in one direction than the other), 95% confidence intervals were calculated from 1000 bootstrap replicates [50].

We used the program BOTTLENECK v1.2.02 [51] to determine whether *H. chrysotricha* populations underwent significant reductions in effective population size (*N*_e_). We used Wilcoxon’s signed rank test, which examines whether populations exhibit a greater level of heterozygosity than predicted in a population at mutation-drift equilibrium, to detect bottlenecks occurring over approximately the last 2–4 *N*_e_ generations. For each *H. chrysotricha* population, we performed 10,000 simulations for each of three mutation models (IAM, infinite allele model; SMM, stepwise mutation model; and TPM, two-phase mutation model, with 95% single-step and 5% multi-step mutations). Statistical significance was set at the 0.05 level.

## 5. Conclusions

Based on high-resolution SNPs, we compared the genetic response of *H. chrysotricha* to short-term and long-term habitat fragmentation. The levels of genetic diversity and the population substructure of *H. chrysotricha* populations were both higher in the TIL region than in the ZA region, while gene flow was higher and less asymmetric among ZA populations. The differences in genetic diversity between island and mainland populations in both regions were not significant, suggesting that genetic erosion of island populations does not increase as the spatial and temporal scales of fragmentation increase. Population connectivity in *H. chrysotricha* has also not been greatly modified by either short-term or long-term habitat fragmentation. The strong fruit (seed) dispersal capacity of *H. chrysotricha*, facilitated by the matrix of water in the island system, could buffer against the negative effects of habitat fragmentation and maintain the population connectivity over thousands of years.

## Figures and Tables

**Figure 1 plants-11-01800-f001:**
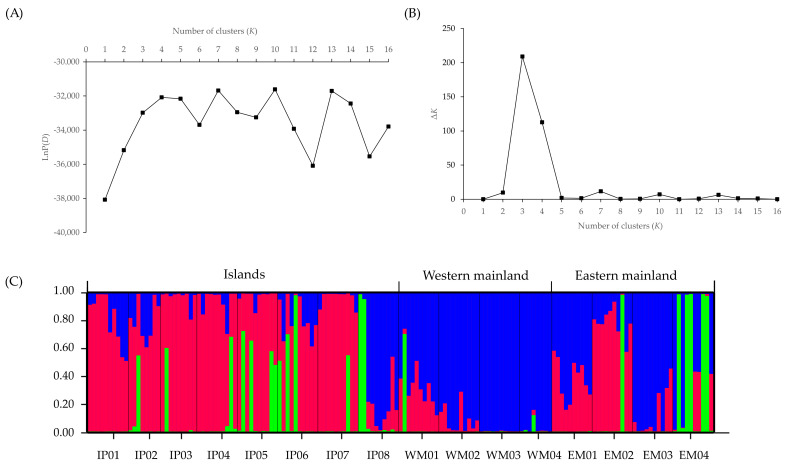
STRUCTURE analysis of 155 individuals (16 populations) of *Hedyotis chrysotricha* from the Thousand-Island Lake (TIL) region, based on RAD-Seq-derived single nucleotide polymorphism (SNP) data. (**A**) Plots of the mean posterior probability [LnP(*D*)] values of each *K* and (**B**) the corresponding Δ*K* statistics. (**C**) Histogram of the STRUCTURE analysis for the model with *K* = 3 (showing the highest Δ*K*). A vertical bar represents a single individual, and each color corresponds to a suggested cluster (Cluster I: dark blue; Cluster II: red; Cluster III: green). The *x*-axis corresponds to population codes. The *y*-axis presents the estimated membership coefficient (*Q*) for each individual in the three clusters.

**Figure 2 plants-11-01800-f002:**
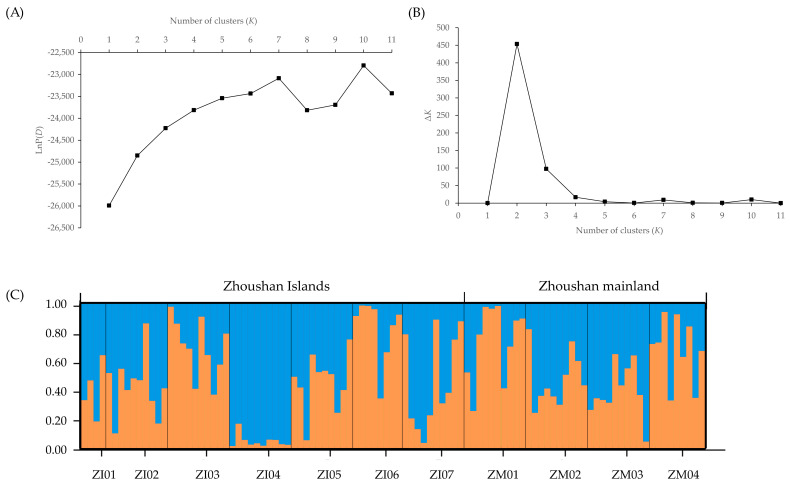
STRUCTURE analysis of 101 individuals (11 populations) of *Hedyotis chrysotricha* from the Zhoushan Archipelago (ZA) region, based on RAD-Seq-derived single nucleotide polymorphism (SNP) data. (**A**) Plots of the mean posterior probability [LnP(*D*)] values of each *K* and (**B**) the corresponding Δ*K* statistics. (**C**) Histogram of the STRUCTURE analysis for the model with *K* = 2 (showing the highest Δ*K*). A vertical bar represents a single individual, and each color corresponds to a suggested cluster (Cluster I: blue; Cluster II: orange). The *x*-axis corresponds to population codes. The *y*-axis presents the estimated membership coefficient (*Q*) for each individual in the two clusters.

**Figure 3 plants-11-01800-f003:**
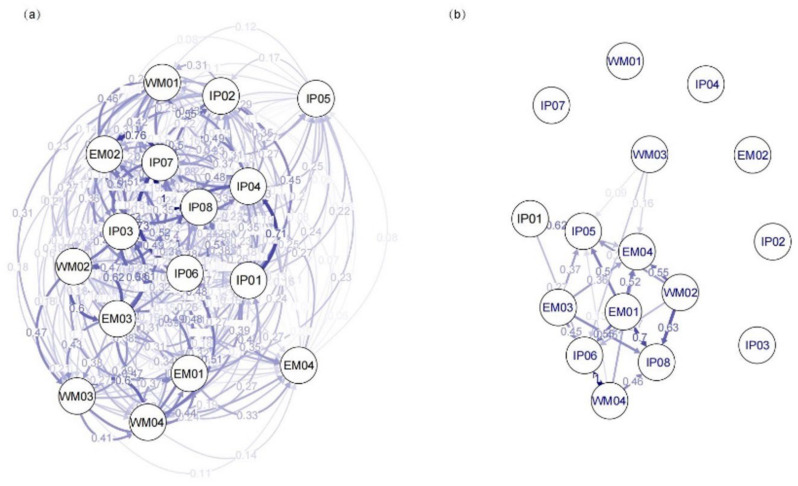
(**a**) The migration network among the 16 *Hedyotis chrysotricha* populations in the Thousand-Island Lake (TIL) region based on *G*_ST_ (coefficient of gene differentiation) values and (**b**) significant relative migration relationships among populations (after 1000 bootstraps). Arrow thickness and color tone are proportional to the magnitude of the gene flow.

**Figure 4 plants-11-01800-f004:**
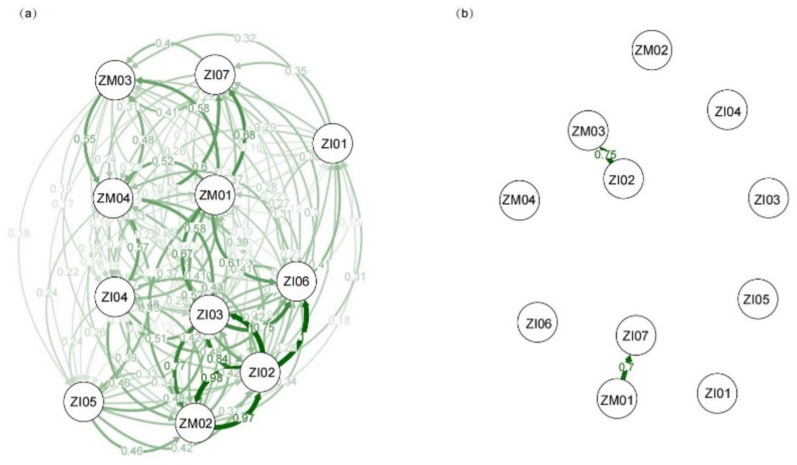
(**a**) The migration network among the 11 *Hedyotis chrysotricha* populations in the Zhoushan archipelago (ZA) region based on *G*_ST_ (coefficient of gene differentiation) values and (**b**) significant relative migration relationships among populations (after 1000 bootstraps). Arrow thickness and color tone are proportional to the magnitude of the gene flow.

**Figure 5 plants-11-01800-f005:**
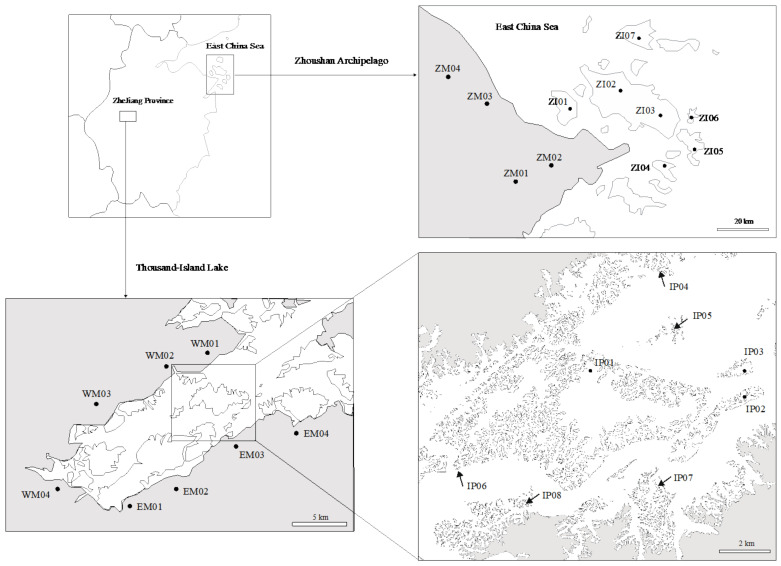
Sample localities of *Hedyotis chrysotricha* in the Thousand-Island Lake (TIL) region and the Zhoushan Archipelago (ZA) region of southeast China. Population codes are identified in Table 1 and Table 2.

**Table 1 plants-11-01800-t001:** Genetic characteristics of island and mainland populations of *Hedyotis chrysotricha* in the Thousand-Island Lake (TIL) region.

TIL	Population ID	Island Size (ha)	Latitude	Longitude	Sample Size	*π*	*H* _O_	*H* _E_	*F* _IS_
Island	IP01	1320	29.5340	118.8818	10	0.236	0.312	0.219	−0.119
	IP02	47.98	29.5254	118.9395	8	0.254	0.316	0.231	−0.080
	IP03	27.49	29.5365	118.9402	9	0.250	0.308	0.231	−0.063
	IP04	1.33	29.5625	118.8990	10	0.255	0.326	0.236	−0.095
	IP05	0.86	29.5487	118.9154	10	0.274	0.317	0.254	−0.053
	IP06	2.56	29.5019	118.8319	10	0.270	0.320	0.250	−0.040
	IP07	2.17	29.4862	118.9098	10	0.252	0.308	0.234	−0.055
	IP08	8.98	29.4949	118.8610	10	0.263	0.294	0.244	−0.008
Mainland	WM01	-	29.5789	118.8671	10	0.240	0.302	0.223	−0.063
	WM02	-	29.5563	118.8154	10	0.227	0.306	0.211	−0.136
	WM03	-	29.4894	118.6907	10	0.221	0.300	0.206	−0.126
	WM04	-	29.4056	118.6367	8	0.219	0.296	0.200	−0.127
	EM01	-	29.3867	118.7484	10	0.235	0.308	0.218	−0.117
	EM02	-	29.4104	118.8441	10	0.256	0.306	0.237	−0.035
	EM03	-	29.4615	118.9056	10	0.221	0.296	0.206	−0.121
	EM04	-	29.4816	119.0097	10	0.275	0.302	0.255	−0.017
Mean (All)						0.247	0.307	0.228	−0.078

Abbreviations: *π*, nucleotide diversity; *H*_O_, observed heterozygosity; *H*_E_, expected heterozygosity; *F*_IS_, within-population inbreeding coefficient.

**Table 2 plants-11-01800-t002:** Genetic characteristics of island and mainland populations of *Hedyotis chrysotricha* in the Zhoushan Archipelago (ZA) region.

ZS	Population ID	Island Size (ha)	Latitude	Longitude	Sample Size	*π*	*H* _O_	*H* _E_	*F* _IS_
Island	ZI01	8635	30.0399	121.8784	4	0.221	0.256	0.181	−0.053
	ZI02	50,265	30.1021	121.1167	10	0.209	0.261	0.194	−0.075
	ZI03	50,265	30.0040	122.2927	10	0.206	0.263	0.191	−0.085
	ZI04	4170	29.8266	122.3036	10	0.208	0.253	0.194	−0.064
	ZI05	6182	29.8573	122.3857	10	0.198	0.250	0.184	−0.068
	ZI06	1185	29.9790	122.3782	8	0.213	0.254	0.194	−0.053
	ZI07	11,512	30.2637	122.1850	10	0.202	0.255	0.187	−0.077
Mainland	ZM01	-	29.8350	121.8706	10	0.195	0.243	0.181	−0.058
	ZM02	-	29.8490	121.7392	10	0.215	0.258	0.200	−0.064
	ZM03	-	30.0183	121.5079	10	0.206	0.257	0.191	−0.074
	ZM04	-	30.0685	121.4286	9	0.213	0.260	0.197	−0.055
Mean (All)						0.208	0.256	0.190	−0.066

Abbreviations: *π*, nucleotide diversity; *H*_O_, observed heterozygosity; *H*_E_, expected heterozygosity; *F*_IS_, within-population inbreeding coefficient.

**Table 3 plants-11-01800-t003:** Pairwise *F*_ST_ values among the 16 populations of *Hedyotis chrysotricha* in the Thousand-Island Lake (TIL) region.

	IP01	IP02	IP03	IP04	IP05	IP06	IP07	IP08	WM01	WM02	WM03	WM04	EM01	EM02	EM03	EM04
IP01																
IP02	0.028															
IP03	0.015	0.090														
IP04	0.022	0.010	0.018													
IP05	0.200	0.006	0.066	**0.185**												
IP06	0.093	0.007	0.006	0.004	0.073											
IP07	0.031	0.029	0.050	0.017	**0.160**	0.031										
IP08	0.092	0.004	0.019	0.008	0.077	0.085	0.001									
WM01	0.047	0.079	0.085	0.002	0.058	0.015	0.019	0.020								
WM02	0.003	0.047	0.008	0.026	0.226	**0.105**	0.013	0.089	0.024							
WM03	0.092	0.002	0.039	0.030	0.201	**0.094**	0.018	0.085	0.047	0.014						
WM04	0.083	0.014	0.024	0.038	0.172	0.004	0.044	0.002	0.051	0.028	0.064					
EM01	0.069	0.047	0.008	0.004	0.211	0.085	0.030	0.083	0.032	0.076	0.049	0.083				
EM02	0.011	0.060	0.079	0.048	0.102	0.056	0.067	0.073	0.049	0.014	0.018	0.058	0.007			
EM03	0.028	0.038	0.012	0.040	0.223	**0.104**	0.034	0.093	0.021	0.030	0.001	0.007	0.021	0.018		
EM04	**0.370**	**0.238**	**0.251**	**0.285**	**0.204**	0.108	**0.287**	0.066	**0.290**	**0.393**	**0.366**	**0.288**	0.376	0.196	**0.386**	

Values in bold are significantly different from zero after sequential Bonferroni correction.

**Table 4 plants-11-01800-t004:** Pairwise *F*_ST_ values among the 11 populations of *Hedyotis chrysotricha* in the Zhoushan Archipelago (ZA) region.

	ZI01	ZI02	ZI03	ZI04	ZI05	ZI06	ZI07	ZM01	ZM02	ZM03	ZM04
ZI01											
ZI02	0.147										
ZI03	0.157	0.075									
ZI04	0.096	0.066	0.013								
ZI05	0.199	**0.303**	**0.270**	0.067							
ZI06	0.162	0.082	0.089	0.057	**0.348**						
ZI07	0.055	0.053	0.036	0.094	**0.315**	0.014					
ZM01	0.139	0.031	0.065	0.053	**0.352**	0.087	0.010				
ZM02	0.014	0.034	0.056	0.078	**0.143**	0.085	**0.174**	**0.158**			
ZM03	0.002	0.059	0.038	**0.167**	**0.436**	0.011	0.051	0.001	**0.242**		
ZM04	0.125	0.073	0.096	0.063	**0.362**	0.123	0.006	0.083	**0.128**	0.001	

Values in bold are significantly different from zero after sequential Bonferroni correction.

**Table 5 plants-11-01800-t005:** Bottleneck analyses for a total of 27 populations of *Hedyotis chrysotricha*, including 16 from the Thousand-Island Lake (TIL) region and 11 from the Zhoushan Archipelago (ZA) region. The *p*-values are shown for the Wilcoxon signed rank tests, as evaluated under the infinite alleles model (IAM), the two-phase mutation model (TPM), and the stepwise mutation model (SMM), respectively.

TIL Region	*p* (Wilcoxon Signed Rank Test)			ZS Region	*p* (Wilcoxon Signed Rank Test)		
Population	IAM	TPM	SMM	Population	IAM	TPM	SMM
IP01	0.691	0.984	0.864	ZI01	**0.000**	**0.001**	**0.007**
IP02	0.949	0.999	0.997	ZI02	**0.028**	0.216	0.915
IP03	0.801	0.995	0.958	ZI03	0.202	0.428	0.977
IP04	0.982	0.999	0.997	ZI04	0.175	0.628	0.977
IP05	**0.000**	**0.003**	**0.000**	ZI05	0.137	0.469	0.958
IP06	0.262	0.973	0.786	ZI06	0.211	0.611	0.966
IP07	0.859	0.999	0.959	ZI07	**0.019**	0.242	0.844
IP08	**0.017**	0.637	0.220	ZM01	0.331	0.694	0.981
WM01	0.992	0.999	0.999	ZM02	**0.021**	0.203	0.835
WM02	0.436	0.989	0.786	ZM03	**0.021**	0.206	0.889
WM03	0.654	0.996	0.860	ZM04	**0.034**	0.331	0.909
WM04	0.942	0.999	0.996				
EM01	0.358	0.983	0.660				
EM02	0.873	0.999	0.981				
EM03	0.744	0.996	0.926				
EM04	**0.000**	**0.001**	**0.000**				

Bold values indicate statistical significance (*p* < 0.05).

## Data Availability

SNPs in VCF files across the Thousand-Island Lake (TIL) and the Zhoushan Archipelago (ZA) populations were deposited in Mendeley Data, V1, https://doi.org/10.17632/wsx8swxd27.1 (accessed on 22 May 2022).

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
