# Peer review of "A Comparative Study of Genetic Responses to Short- and Long-Term Habitat Fragmentation in a Distylous Herb Hedyotis chyrsotricha (Rubiaceae)"

_plants, 2022, doi:10.3390/plants11141800_

Round 1
Reviewer 1 Report
MAJOR COMMENTS
- Fig.2 (A) Looking at plot A, it appears that the Delta K value continues to rise after K=2. Did the authors also observe K>2?
- The discussion is shallow. In this section, results are reported and discussed, but comparisons with other similar studies are lacking.
MINOR COMMENTS
- l.50 enter reference number
- l.73 delete "which"
- l.77 enter reference number
-Tab.3 enlarge the table columns or write in a smaller font; write the caption in the same font as Tab.4
- l.161 correct "..and Cluster II (Figure 2)" with "..and Cluster II (Figure 1)"
- Fig.3 and Fig.4: if it is possible, it would be better to make bigger pictures to better see the values of migration relationships
- l.252 change "maintain" with "maintaining"
Reviewer 2 Report
In the Table 3 please explain the meaning of numbers given under Fst values.
Reviewer 3 Report
I enjoyed reading the manuscript on the population genetics of Hedyotis chrysotricha. The authors did a nice job of introducing the topic and explaining the methods focused on multiple aspects of the population genetics of the species in two regions with multiple sampled populations. The authors resolved that while genetic variation exists, genetic diversity was not necessarily high across the populations of the two areas studied. Additionally, putative gene flow is occurring to result in some homogeneity of the populations.
I was left with a few thoughts after reading the manuscript (see below), and I have included comments below. One overall comment is that while the species studied is distylous, there is little mention of this breeding system and its role in influencing population structure in the species. That could be worth exploring more, especially as the discussion is not particularly extensive.
-Castanopsis sclerophylla is Castanopsis sclerophylla (Lindl. & Paxton) Schottky
-In the introduction, it could be worthwhile to introduce aspects of the population genetics of other distylous plants
-Were both LS and SS individuals sampled for the study?
-Was there any missing data in the matrix?
-Why wasn't the STRUCTURE analysis also run without admixture?
-How do the divMigrate results compare to those from BAYESASS? Also, I think that divMigrate was developed for microsatellite loci. Do you have a sense as to how well it would work for SNPs?
-Including the graphs for K values was useful. Are there other values of K that might be helpful that show the STRUCTURE bar graphs in order to provide additional information on population structure?
-The number of SNPs employed in the present study appears to be less than other, similar types of studies. How might that have impacted the results?
-"Besides, some of the few species of Hedyotis with indehiscent fruits possess a fleshy mesocarp, which may provide the possibility of fruit dispersal by water as well [14]." But is this the case for H. chrysotricha? If not, it's likely not relevant to this particular discussion. Additional details on the the influence of water on wind-dispersed seeds would be useful to include, especially if there is particular evidence that water dispersal, not wind dispersal, plays a role in patterns of migration in this species.
-It would be nice to see a greater discussion on the reason for genetic similarity, especially giving that there was not necessarily strong gene flow among the islands and that there may have been a genetic bottleneck in multiple populations, both of which should increase genetic isolation, and, therefore, population structure (at least in my understanding). Is the LS-SS ratio similar among the populations? A negative Fis suggests that there's a fair amount of inbreeding, yet the populations, especially in ZA, appear pretty homogenous.
-Why is EM04 so genetically different from the other populations?
-In the abstract, I do not believe there was profound population substructure. There seems to be three population clusters, which does not strike me as profound.
Author Response
Thank you for your careful check. The results were not clearly displayed due to a table layout problem. We have already readjusted this table.Round 2
Reviewer 3 Report
I enjoyed reading the updated version of the manuscript on the population genetics of Hedyotis. Unfortunately, I couldn't see the authors responses to my prior comments (therefore, I don't know how some of the comments that weren't changed in the manuscript were addressed), but multiple aspects of the manuscript have been updated and improved. I believe that some parts of the discussion could continue to be strengthened and addressed, and I have listed them below.
-line 89: Optimal populations is an unusual phrase. I'd suggest natural populations
-line 93: herkogamy not hierogamy, also the morphs don't necessarily differ in their reproductive success, but tend to have various self- and intramorph incompatibility systems
-Line 267: What is the foraging distance of the pollinators of this species of Hedyotis? Could a stepping-stone model help explain the genetic structure? Addressing this could help make the other statement regarding hydrochory more plausible because, as is, it still seems speculative given the lack of understanding of diaspore dispersal of this species, as the authors themselves state. I would think that the results of BAYESASS and divMigration could provide additional information as well concerning the amount of genetic exchange that could strengthen the discussion.
-line 308: It could be useful to conduct isolation-by-distance analyses to get a better understanding of the role of isolation in the the two geographic areas.
-line 377: It seems like caution is still warranted with divMigrate in general and with SNP data. How did the results of BAYESASS compare with those of divMigrate?
